# Glutamine as an Anti-Fatigue Amino Acid in Sports Nutrition

**DOI:** 10.3390/nu11040863

**Published:** 2019-04-17

**Authors:** Audrey Yule Coqueiro, Marcelo Macedo Rogero, Julio Tirapegui

**Affiliations:** 1Department of Food and Experimental Nutrition, Faculty of Pharmaceutical Sciences, University of São Paulo, Avenida Professor Lineu Prestes 580, São Paulo 05508-000, Brazil; tirapegu@usp.br; 2Department of Nutrition, Faculty of Public Health, University of São Paulo, Avenida Doutor Arnaldo 715, São Paulo 01246-904, Brazil; mmrogero@usp.br; 3Food Research Center (FoRC), CEPID-FAPESP, Research Innovation and Dissemination Centers São Paulo Research Foundation, São Paulo 05468-140, Brazil

**Keywords:** amino acid, muscle fatigue, central fatigue, performance, immune system, hydration

## Abstract

Glutamine is a conditionally essential amino acid widely used in sports nutrition, especially because of its immunomodulatory role. Notwithstanding, glutamine plays several other biological functions, such as cell proliferation, energy production, glycogenesis, ammonia buffering, maintenance of the acid-base balance, among others. Thus, this amino acid began to be investigated in sports nutrition beyond its effect on the immune system, attributing to glutamine various properties, such as an anti-fatigue role. Considering that the ergogenic potential of this amino acid is still not completely known, this review aimed to address the main properties by which glutamine could delay fatigue, as well as the effects of glutamine supplementation, alone or associated with other nutrients, on fatigue markers and performance in the context of physical exercise. PubMed database was selected to examine the literature, using the keywords combination “glutamine” and “fatigue”. Fifty-five studies met the inclusion criteria and were evaluated in this integrative literature review. Most of the studies evaluated observed that glutamine supplementation improved some fatigue markers, such as increased glycogen synthesis and reduced ammonia accumulation, but this intervention did not increase physical performance. Thus, despite improving some fatigue parameters, glutamine supplementation seems to have limited effects on performance.

## 1. Introduction

Fatigue is defined as the inability to maintain power output and strength, impairing physical performance [1]. The main causes of fatigue are: accumulation of protons in the muscle cell, depletion of energy sources (e.g., phosphocreatine and glycogen), accumulation of ammonia in the blood and tissues [2,3,4], oxidative stress, muscle damage [1] and changes in neurotransmitter synthesis, such as the increase in serotonin and the decrease in dopamine [5].

In order to delay the onset of fatigue and to improve athletic performance, several nutritional strategies have been applied. Since the mid-1980s and 1990s, the role of amino acids in the development of fatigue has been discussed [3,6,7,8,9], and evidence demonstrated that plasma glutamine concentrations and the glutamine/glutamate plasma ratio are reduced in athletes under chronic fatigue and overtraining syndrome, raising a question about the possible ergogenic effects of glutamine supplementation [10,11,12,13].

Glutamine could delay fatigue by several mechanisms: (i) it is one of the most abundant glycogenic amino acids in humans and animals, having a significant influence on the anaplerosis of the Krebs cycle and gluconeogenesis [14,15], (ii) through the activation of glycogen synthase, glutamine is considered as a direct stimulator of glycogen synthesis [7,16], (iii) this amino acid is the main non-toxic ammonia carrier, avoiding the accumulation of this metabolite [14], (iv) glutamine is also linked to attenuation of muscle damage and is considered an indirect antioxidant via stimulation of glutathione synthesis [17,18], among others.

Despite the potential of glutamine in attenuating some causes of fatigue, the effects of this amino acid supplementation on fatigue markers and physical performance have not been yet completely elucidated. Thus, the present article aims to review the main anti-fatigue properties of glutamine and the effects of this amino acid supplementation in this regard.

## 2. Methods

The integrative literature review method was based on the five stages (problem identification, literature search, data evaluation, data analysis and presentation) proposed by Whittemore and Knafl [19] and the improvement of this method proposed by Hopia et al. [20].

### 2.1. Problem Identification

The purposes of this article were to review the main anti-fatigue properties of glutamine and to critically analyze the literature regarding the effects of glutamine supplementation (alone or with other nutrients) on exercise-induced fatigue in healthy animals and humans.

### 2.2. Literature Search

PubMed database was selected to examine the literature, in February 2019, using the descriptor Medical Subject Headings (MeSH), without limitation as to the publication period. The keyword combination used was “Glutamine” and “Fatigue” (*n* = 122 articles).

Articles discussing fatigue related to diseases or that included animals or humans with any stated medical condition were excluded from this study. Only the articles that addressed the relationship between glutamine and fatigue induced by physical exercise in healthy individuals were included in this review. Additionally, unpublished manuscripts (such as dissertations and thesis) were not included in this study.

### 2.3. Data Extraction

One hundred and twenty-two articles were found. After reading the title of these studies, 61 articles were excluded, since they had no correlation with the subject (effects of glutamine supplementation on exercise-induced fatigue) or did not provide the complete version of the manuscript (just the abstract). Of the 61 articles that remained, 19 articles were excluded after reading the abstract, since they did not have a correlation with the theme, remaining 42 studies.

After reading the complete version of these 42 selected articles, 13 other studies, which were cited in the articles evaluated, but were not obtained in the search, were included, totalizing 55 articles—44 original studies and 11 literature reviews (Figure 1).

### 2.4. Data Synthesis

Fifty-five articles, which evaluated and/or discussed glutamine supplementation, alone or associated with other nutrients, in the context of fatigue induced by physical exercise, were included in this review.

Concerning the animal and human studies, the aspects of all of these articles were described thoroughly. Certain features of these studies, such as author, participants, study design and findings were described in tables. Moreover, the limitations of these studies were discussed.

## 3. Glutamine and Physical Exercise

Glutamine is a five-carbon neutral amino acid, containing the molecular weight of 146.15 g/mol, and considered as the most abundant free amino acid in the human body [15]. In adult humans following overnight fast, the normal blood levels of glutamine are 550–750 µmol/L [21], contributing for more than 20% of the blood amino acid pool [22]. In the skeletal muscle, glutamine comprises 50–60% of the total free amino acid pool, being considered as the most synthesized amino acid in the human muscle, especially in slow-twitch muscles, which contain glutamine concentrations 3-fold higher than fast-twitch muscles [22,23]. Therefore, skeletal muscle releases glutamine into the circulation at high rates, approximately 50 mmol per hour in the fed state [21].

Organs can be classified as producers or consumers of glutamine – skeletal muscles, lungs, liver, brain and adipose tissue present high activity of glutamine synthetase (an enzyme that synthesizes glutamine from ammonia and glutamate in the presence of adenosine triphosphate-ATP) and are considered as glutamine producers. On the other hand, leucocytes, enterocytes, colonocytes, thymocytes, fibroblasts, endothelial cells, and kidney tubular cells present high activity of glutaminase (an enzyme that hydrolyzes glutamine, converting it into glutamate and ammonia) and are classified as consumers of glutamine [2,24,25,26,27,28].

Glutamine is involved in several biological functions, such as nucleotide synthesis, cell proliferation, regulation of protein synthesis and degradation, energy production, glycogenesis, ammonia detox, maintenance of the acid-base balance, among others. Moreover, this amino acid regulates the expression of several genes associated with metabolism and activates many intracellular signaling pathways [15]. Nutritionally, glutamine is considered as conditionally essential, since in catabolic situations, such as clinical traumas, burns, sepsis, and prolonged and exhaustive exercises, the endogenous synthesis of glutamine might not be sufficient to supply the body demand, and glutamine deficiency can occur [24,25].

Since the mid-1970s and 1980s, glutamine metabolism has been investigated during and after physical exercise [8], and it was observed that blood glutamine responds differently according to the duration of the exercise [2]. Short-term exercise increases muscle release of glutamine and its blood concentrations [4], whereas, in long-term and exhaustive exercises, such as marathon race, muscle synthesis of glutamine is insufficient to meet the body need for this amino acid, decreasing blood glutamine [11,16,29,30,31]. This decrease is transient and seems to last for 6–9 h after a marathon [24], and is accompanied by a 30–40% fall in muscle glutamine or in its precursors, like glutamate [11]. Nevertheless, it is worth mentioning that some studies demonstrated that even after exhaustive exercises (ultra-triathlon), blood glutamine did not change [6].

Decreased glutamine availability is linked with disorders in the immune system and an increase in the incidence of infections [24,25]. Santos et al. [32] observed, in an experimental model (rats), that exhaustive exercise induces an increase in macrophages functionality (phagocytosis and H_2_O_2_ production), as well as an augment in glutamine consumption and metabolism in these cells, indicating the importance of glutamine for macrophages functionality in the post-training period and suggesting a possible role for glutamine supplementation to individuals involved in exhaustive exercises [32].

Regarding the glutamine supplementation, evidence indicates that plasma glutamine, in response to glutamine supplementation, increases markedly within 30 min after supplementation, returning to basal levels about 2 h after glutamine administration [29]. Moreover, doses of 20–30 g of glutamine have been reported to be tolerated (no side effects), offering no harm to humans [21].

Initially, glutamine was supplemented mainly because of its immunomodulatory potential [24]. However, since this amino acid plays a wide variety of biological activities, glutamine began to be investigated in sports nutrition beyond its effect on the immune system, attributing to this amino acid several properties, such as an anti-fatigue role.

## 4. Glutamine and Its Anti-Fatigue Properties

Fatigue is a multiple-cause phenomenon defined as the inability to maintain power output and strength, resulting in physical and mental performance impairment. Conceptually, fatigue may be classified as peripheral, also called muscle fatigue, when the biochemical changes occur within the skeletal muscle cell, or central, comprising disturbances in the central nervous system (CNS) that limit performance [1].

The main causes of fatigue are: (i) accumulation of protons in the muscle cell, reducing the pH and affecting the activity of enzymes, such as phosphofructokinase, (ii) depletion of energy sources (e.g., phosphocreatine and glycogen) for the continuity of the exercise, (iii) accumulation of ammonia (toxic metabolite) in the blood and tissues [2,3,4], (iv) oxidative stress, (v) muscle damage [1] and (vi) changes in neurotransmitter synthesis, such as the increase in serotonin and the decrease in dopamine [5], which may cause a state of tiredness, sleep, and lethargy during prolonged exercises [33].

The underlying mechanisms behind the increase in brain serotonin are the plasma increase in its precursor, free (not albumin-bound) tryptophan, and the plasma decrease in the large neutral amino acids, such as branched-chain amino acids (BCAA), which compete with tryptophan to enter the brain. In addition, during long-term exercise, the augment in free fatty acid (FFA) concentrations can displace tryptophan from albumin, increasing free tryptophan and facilitating its brain influx and, consequently, serotonin synthesis [33].

Regardless of the origin (peripheral or central), fatigue is a complex and multifaceted phenomenon, since several factors may limit performance, but the improvement of single markers might not necessarily delay fatigue. Furthermore, it is worth highlighting that some causes of fatigue are not completely elucidated in the literature, such as the relationship between increased serotonin synthesis and performance decrease [1,33].

In order to delay the onset of fatigue and to improve athletic performance, several nutritional strategies are applied. Since the mid-1980s and 1990s, the role of amino acids in the development of fatigue has been discussed [3,6,7,8,9], and evidence demonstrated that blood glutamine and the glutamine/glutamate blood ratio were reduced after strenuous exercises [2,11,12,13,34,35,36], although some studies did not corroborate these findings [3,6].

Jin et al. [10] observed a drastic decrease in plasma, muscle, and liver glutamine concentrations in an animal model of complex fatigue (forced swimming). Similarly, Kingsbury et al. [11] verified that elite athletes under chronic fatigue (for several weeks) presented critical concentrations of blood glutamine (<450 µmol/L) and higher prevalence of infections compared to athletes without fatigue. An increase in protein intake (through lean meat, fish, cheese, milk powder and soya, that is, glutamine-rich foods) to these fatigued athletes enhanced blood glutamine levels and improved physical performance, raising the question about the possible anti-fatigue effects of glutamine supplementation [29].

Glutamine is one of the most abundant glycogenic amino acids in humans and animals, having a significant influence on the anaplerosis of the Krebs cycle and gluconeogenesis, being the most important energy substrate for renal gluconeogenesis [14,15]. Additionally, glutamine is a direct stimulator of glycogen synthesis via the activation of glycogen synthetase, possibly through a mechanism of cell-swelling and to the diversion of glutamine carbon to glycogen, increasing hepatic and muscle glycogen stores [7,16,33].

Glutamine is also associated with the prevention of ammonia accumulation. Ammonia production during exercise occurs via amino acid oxidation and in energy metabolism (adenosine monophosphate-AMP deamination), indicating the reduction of ATP concentration and glycogen content [1]; thus, glutamine supplementation could minimize ammonia production due to its effects on energy metabolism [14]. Ammonia accumulation is an important cause of fatigue since this metabolite is toxic and affects the activity of some flux-generating enzymes, the cell permeability to ions and the ratio of NAD^+^/NADH [37]. However, as a consequence of the increase in ammonia production during exercise, glutamine synthesis is augmented, as a mechanism of ammonia buffering [37].

Guezennec et al. [9] observed an increase in blood and brain ammonia in rats after running until exhaustion, followed by an enhance in brain glutamine and a decrease in brain glutamate. Based on these data, the authors concluded that the increase in brain ammonia levels stimulates glutamine synthesis as a mechanism of detoxication. Corroborating these results, Blomstrand et al. [38] verified an increase in the brain release of glutamine during an exhaustive exercise (3 h in the cycle ergometer), suggesting that the increase in glutamine synthesis in the brain, as a mechanism of ammonia buffering, results in a higher brain release of glutamine.

Glutamine may also attenuate ammonia accumulation because this amino acid is the main transporter of nitrogen (ammonia) in the body, preventing the muscle accumulation of this metabolite, and favoring ammonia hepatic metabolism, as well as its renal excretion [14,33].

Muscle damage and oxidative stress are other causes of fatigue that could be minimized by glutamine. Studies in our laboratory showed that glutamine supplementation (for 21 days) reduced the plasma concentrations of creatine kinase (CK) and lactate dehydrogenase (LDH)—markers of muscle damage—in rats submitted to strenuous resistance training [17,18]. Several mechanisms might explain this protective effect of glutamine; this amino acid is absorbed through a sodium-dependent transport, increasing the intracellular concentration of sodium ions and promoting water retention, which increases cell hydration and its resistance to lesions [17]. Glutamine also presents an important immunomodulatory role, increasing the synthesis of anti-inflammatory and cytoprotective factors, such as interleukin 10 (IL-10) and heat-shock protein (HSP) [17].

Moreover, evidence indicates that glutamine is an important donator of glutamate for glutathione synthesis—the most important non-enzymatic antioxidant in the cell—which may indicate an indirect antioxidant effect of glutamine [18]. Although elevated oxidative stress might contribute to fatigue, it is unclear in the literature whether the increase in glutathione concentrations through glutamine supplementation could attenuate fatigue and improve physical performance. It is important mentioning that some of these results (attenuation of muscle damage and oxidative stress parameters) were obtained from animal studies, thus, it is not possible to guarantee that the same effects would occur in human trials. In addition, recent position stands of well-recognized organizations, such as the International Society of Sports Nutrition (ISSN) and the International Olympic Committee (IOC), have considered glutamine as a non-effective supplement, with little or no evidence of efficacy [39,40].

Finally, another possible anti-fatigue property of glutamine is to prevent dehydration. Glutamine is transported across the intestinal brush border by a sodium-dependent system, promoting more rapid fluid and electrolyte absorption in the gut. Therefore, the inclusion of glutamine in rehydration solutions might increase sodium absorption and bulk water flow [7,41]. When glutamine is administered with alanine, as dipeptide (L-alanyl-L-glutamine), the fluid and electrolyte absorption seem to be even higher than supplementation with glutamine alone since dipeptide presents great stability in solution and low pH [41]. Considering the potential properties presented, glutamine seems to be an interesting supplement for fatigue attenuation, especially for athletes who practice endurance sports (exhaustive and prolonged exercise). In Figure 2, the main properties of glutamine in delaying fatigue are presented.

### 4.1. Effects of Glutamine Supplementation on Exercise-Induced Fatigue Glutamine

The effects of glutamine infusion after an exhaustive exercise (cycling at 70–140% of the VO_2max_ for 90 min) were first tested in 1995. Three groups of individuals were submitted to exercise and infusion (30 min after completing the exercise) of (i) glutamine, (ii) alanine and glycine or (iii) saline. Muscle glutamine concentrations were increased during glutamine infusion, reduced during alanine and glycine infusion and remained constant during saline infusion. Two hours after exercise, the muscle glycogen content was higher in the subjects treated with glutamine compared to other groups. This study suggested that glutamine has effects on glycogen synthesis beyond its gluconeogenic role, since alanine and glycine, despite providing glucose through gluconeogenesis, did not affect muscle glycogen [16].

Similarly, Bowtell et al. [7] investigated the effects of glutamine supplementation on whole body carbohydrate storage and muscle glycogen resynthesis in subjects after completing a glycogen-depleting exercise protocol. Individuals cycled on the ergometer at 70% of the VO_2max_ for 30 min; thereafter, the workload was doubled and they completed 6 times of 1 min bursts of activity separated by 2 min of rest. Finally, they cycled for 45 min at 70% of the VO_2max_. After exercise, individuals received one of the three drinks: (i) 18.5% glucose polymer solution, (ii) 18.5% glucose polymer solution containing 8 g of glutamine or (iii) a placebo containing 8 g of glutamine. Plasma glucose and insulin were higher when consuming drinks with glucose, and there was a tendency for plasma insulin to be higher after ingesting glucose and glutamine rather than only glucose. Supplementation with glutamine-containing drinks increased plasma glutamine. In the second hour of recovery, glucose and glutamine solution increased whole body nonoxidative glucose disposal by 25%, whereas oral glutamine alone promoted the storage of muscle glycogen to an extent similar to glucose. This result is surprising since it would be expected that the provision of 61 g of glucose polymer (amount of glucose provided in the glucose polymer solution), as opposed to 8 g of glutamine (amount of glutamine provided in the placebo solution), would result in a higher muscle glycogen synthesis; thus, it suggests a great impact of glutamine on muscle glycogen synthesis. However, there is limited evidence concerning this effect on glycogen synthesis in the athlete population.

The same research group, in 2001, observed a significant increase in the muscle concentrations of Krebs cycle intermediates, such as citrate, malate, fumarate and succinate, at the beginning of the exercise (bicycle exercise at 70% of the VO_2max_) after acute glutamine supplementation, when in comparison with ornithine α-ketoglutarate or placebo administration. Nonetheless, glutamine supplementation did not affect the extent of phosphocreatine depletion, lactate accumulation or endurance time, suggesting that the muscle concentration of Krebs cycle intermediates was not limiting for energy production and physical performance [42].

Contrary to the abovementioned studies, van Hall et al. [43] verified that the supplementation with free glutamine or a carbohydrate mixture containing glutamine did not affect muscle glycogen resynthesis after exercise. Individuals were submitted to an intense cycle ergometer exercise in order to deplete glycogen. Thereafter, subjects ingested four different drinks in three 500 mL boluses, immediately after exercise, 1 h after exercise and 2 h after exercise. The drinks were: 1—control: 0.8 g/kg of glucose, 2—glutamine: 0.8 g/kg of glucose plus 0.3 g/kg of glutamine, 3—a wheat hydrolysate containing 0.8 g/kg of glucose and 26% of glutamine, and 4—a whey hydrolysate containing 0.8 g/kg of glucose and 6.6% of glutamine. Plasma glutamine was reduced with control drink intake, remained unchanged with hydrolysates (wheat and whey) consumption and was 2-fold increased after glutamine supplementation. Despite increasing plasma glutamine, this amino acid administration did not improve glycogen synthesis rate. The different supplementation protocols and administered doses might explain the differences in the results of these studies.

Besides depleted glycogen stores, other markers of fatigue, such as blood ammonia and muscle damage parameters, were investigated after glutamine supplementation. Carvalho-Peixoto et al. [44] supplemented glutamine and/or carbohydrate for high-trained runners before running for 120 min (~34 km), and observed that, contrary to placebo, there was no increase in blood ammonia in supplemented individuals in the first 30 min of exercise. Additionally, in the last 90 min of running, subjects under all supplementations had lower blood ammonia levels compared to placebo. There was no difference between supplementations, suggesting that glutamine and carbohydrate may attenuate ammonia increase during exercise, but without synergy between them.

Likewise, the effects of glutamine or alanine supplementation, either for short-term (1 day) or long-term (5 days), were investigated on blood ammonia of professional football players after two different exercise protocols—intermittent (a football match) or with continuous intensity (running for 60 min at 80% of the maximum heart rate-HR_max_). Both exercises increased blood ammonia, whereas long-term glutamine supplementation protected against hyperammonemia only after the intermittent exercise, suggesting that the effect of glutamine administration on blood ammonia depends on the duration of the supplementation and on the type of physical exercise [14].

Different from these studies, Koo et al. [45] compared the supplementation with glutamine, BCAA or placebo to elite rowing athletes that were engaged in a session of rowing (2000 m) at the maximal intensity, and observed that none of the interventions affected plasma ammonia, lactate and the cytokines IL-6 and IL-8; nevertheless, glutamine supplementation reduced the plasma levels of CK 30 min after exercise compared to the values measured immediately after training, suggesting a possible effect of glutamine in attenuating muscle damage.

Concerning the physical performance, Favano et al. [46] supplemented glutamine peptide and carbohydrate or only carbohydrate to soccer players that were submitted to an intermittent exercise on the treadmill, and observed an increase in the time and distance (21% and 22%, respectively) and reduced rate of perceived exertion (RPE) after supplementing with glutamine and carbohydrate compared to the administration of only carbohydrate. Similarly, the supplementation with glutamine and carbohydrate to subjects that performed a running-based anaerobic sprint test (6 × 35 m discontinuous sprints) increased the maximal and minimal power compared to placebo (water + sweetener) [47]. Nava et al. [48] also observed that glutamine supplementation reduced subjective fatigue, ratings of perceived exertion and gastrointestinal damage (measured by intestinal fatty acid binding proteins), besides increasing HSP70 and inhibitor of kappa B (IκBα) in peripheral blood mononuclear cells (PBMCs), in individuals submitted to a simulated wildland firefighting session in hot conditions.

In contrast to these studies, Krieger et al. [49] verified that chronic glutamine supplementation did not improve performance during interval training. These data suggest that the combination of glutamine and carbohydrate is more efficient in preventing anaerobic power decrease and increasing performance than glutamine alone, emphasizing the synergy between glutamine and carbohydrate, although some studies did not corroborate this finding.

### 4.2. L-Alanyl-L-glutamine

A high proportion of dietary glutamine is retained in intestinal cells, leaving only small concentrations of glutamine to enter into the bloodstream [29]. In order to increase glutamine availability, the supplementation with peptides of glutamine, such as the dipeptide L-alanyl-L-glutamine, has been used, since di- and tripeptides are absorbed across the intestinal epithelium in their intact form by more efficient and faster mechanisms, such as the oligopeptide transporter PepT-1, than free amino acids [17,18,33]. Thus, evidence showed that L-alanyl-L-glutamine supplementation was more effective in increasing plasma, muscle and liver glutamine concentrations compared to free glutamine administration [50]. Furthermore, L-alanyl-L-glutamine presents higher stability in solution and low pH than glutamine and is a better option to be included in commercial products, such as sports drinks [41].

Rogero et al. [50] supplemented glutamine (GLN) or L-alanyl-L-glutamine (DIP) for 21 days to rats submitted to swimming exercise for 6 weeks, followed by an exhaustion test. Animals were sacrificed immediately after the test (EXA) or after 3 h (REC). Muscle glutamine concentration was higher in DIP-EXA animals compared to CON-EXA and GLN-EXA groups, whereas DIP-REC group presented a higher plasma and liver content of glutamine than CON-REC group. Notwithstanding, muscle glutamine and protein levels were higher in GLN-REC and DIP-REC animals compared to CON-REC. Although supplementations, especially with L-alanyl-L-glutamine, increased glutamine concentrations, there were no differences between groups in the time to exhaustion, indicating that neither glutamine nor L-alanyl-L-glutamine supplementation improved physical performance.

Hoffman et al. [51] administered L-alanyl-L-glutamine, in two doses (0.05 g/kg or 0.2 g/kg), or water to dehydrated male subjects (mild dehydration) submitted to an exercise session on the cycle ergometer at 75% of the VO_2max_, and verified an increase in blood glutamine concentrations with the higher dose of dipeptide, as well as an increase in the time until exhaustion in both groups treated with L-alanyl-L-glutamine compared to water. There was no difference between trials in the parameters of muscle damage (blood CK), inflammation (blood IL-6), oxidative stress (blood malondialdehyde), among others. The authors attributed the performance improvement induced by L-alanyl-L-glutamine supplementation to the possible increase in fluid and electrolyte absorption promoted by this dipeptide; nonetheless, as seen previously, glutamine could delay fatigue through several other mechanisms, such as protecting against hyperammonemia—a parameter that was not measured in this study.

The same research group investigated the effects of L-alanyl-L-glutamine, either in low (1 g/500 mL) or high dose (2 g/500 mL), on physical performance during a basketball game (jump power, reaction time, shooting accuracy and fatigue), and observed an improvement in basketball shooting performance and visual reaction time with low dose of L-alanyl-L-glutamine compared to water ingestion (placebo) [41]. Similarly, McCormack et al. [52] submitted endurance-trained men to a one-hour treadmill run at 75% of the VO_2peak_ followed by a run to exhaustion at 90% of the VO_2peak_, after supplementing them with (i) L-alanyl-L-glutamine and a sports drink, (ii) only the sports drink (placebo) or (iii) without any supplementation (no hydration trial). The authors observed that plasma glutamine was higher and time to exhaustion was longer when supplementing with dipeptide compared to the no hydration trial, but there was no difference between L-alanyl-L-glutamine supplementation and the sports drink only (placebo).

Our research group also investigated the effects of glutamine and alanine supplementation, as dipeptide (L-alanyl-L-glutamine) or in their free form, to rats submitted to a resistance training protocol, consisting of climbing a vertical ladder with progressive loads. We observed that these interventions reduced parameters of muscle damage (plasma CK and LDH) and inflammation (plasma IL-1β and tumor necrosis factor alpha—TNF-α), and increased anti-inflammatory and cytoprotective markers (plasma IL-6, IL-10 and muscle HSP70) [17]. Additionally, these supplementations reduced the oxidized glutathione (GSSG)/reduced glutathione (GSH) ratio in erythrocytes and muscle thiobarbituric acid reactive substances (TBARS), evidencing an antioxidant role [18]. Despite improving several parameters, glutamine and alanine administration did not improve performance evaluated by a maximum carrying capacity test [17,18].

In fact, recently, we observed that these amino acids supplementation improved some fatigue markers, such as muscle ammonia and glycogen, while impaired others, since L-alanyl-L-glutamine administration increased the hypothalamic concentrations of serotonin and the plasma concentrations of its precursor (tryptophan), although without affecting physical performance. It is worth mentioning that serotonin is considered as a parameter of central fatigue, since it is linked to behavioral alterations, such as reduced appetite, sleepiness, and fatigue, reducing the mental and physical efficiency [33]. As previously mentioned, fatigue is a complex phenomenon and the improvement or impairment of single markers may not necessarily affect performance [1].

### 4.3. Glutamine Associated with Other Nutrients

Studies also have evaluated the effects of glutamine, associated with several other amino acids, on fatigue markers. Ohtani et al. [23] observed that an amino acid mixture (glutamine: 0.65 g—the amino acid in the highest concentration in the mixture—leucine, isoleucine, valine, arginine, threonine, lysine, proline, methionine, histidine, phenylalanine and tryptophan), when supplemented for 90 days to elite rugby players, improved reported vigor and earlier recovery from fatigue. Moreover, amino acids administration increased parameters of oxygen-carrying capacity, such as hemoglobin, red blood cells count, hematocrit and serum iron. After one year without the supplementation, all parameters returned to basal values, indicating the need for daily supplementation to maintain the effects. Some limitations of this study should be highlighted. Firstly, since several amino acids were ingested, it is not possible to attribute the effects to any of them, and, secondly, some of the results (such as reported vigor) were obtained by questionnaires. Thus, several factors could have affected the accuracy of the results.

The same research group, in the same year, evaluated this amino acid mixture for middle- and long-distance runners. The athletes were engaged in sustained exercise (running) for 2–3 h/day, 5 days/week, for 6 months. During this period, subjects received three 1-month treatments, separated by one month of washout. The treatments consisted of three different doses of the amino acid mixture: 2.2 g/day, 4.4/day and 6.6 g/day. The main effects were observed with the higher dose (6.6 g/day), which increased the physical condition score and markers of oxygen-carrying capacity (hematocrit, hemoglobin, and red blood cells count), while decreased serum CK, a marker of muscle damage and inflammation [53].

This amino acid mixture was also investigated on the recovery from muscle fatigue following eccentric exercise. Individuals were submitted to a session of eccentric training and, thereafter, they were allowed to recover for 10 days while supplementing with amino acids mixture or placebo. Measures of muscle strength (maximum isometric strength, maximum concentric strength, and maximum eccentric strength) in both elbow flexor and extensor muscles showed an earlier recovery from muscle fatigue when supplementing with amino acids compared to placebo. Additionally, maximum isometric strength was higher in amino acid trials than in the placebo, and most of the individuals reported less delayed muscle soreness with amino acids supplementation, indicating an ergogenic effect of this intervention [54].

Likewise, Willems et al. [55] tested the supplement ‘Cyclone^TM^’, which contains whey protein (30 g), glutamine (5.1 g), creatine (5.1 g) and β-hydroxy-β-methylbutyrate (HMB) (1.5 g), for subjects submitted to 12 weeks of resistance training, and observed that this intervention improved some performance parameters, such as the number of repetitions for 80% pre-training 1-RM for lateral pull and bench press, but not others, like maximum voluntary isometric force (MVIF), time to fatigue at 70% of the MVIF, peak concentric strength and 1-RM of lateral pull. The authors concluded that this multi-ingredient supplement improves the ability to perform some resistance training-specific tasks.

Corroborating these data, an interesting study observed that the voluntary intake of a solution containing BCAA (15.2 mmol/L of leucine, 9.9 mmol/L of isoleucine, 11.1 mmol/L of valine), glutamine (16.6 mmol/L), and arginine (13.9 mmol/L), rather than water, was positively correlated with timing and volume of exercise in rats exercised on running wheels, indicating a preference for this amino acid solution as a consequence of the exercise practice. In addition, the intake of these amino acids increased the BCAA/tryptophan plasma ratio and decreased the brain release of serotonin, a central fatigue parameter [5].

In opposition to the studies abovementioned, Kersick et al. [56] did not verified any effect of a supplementation containing whey protein (40 g), glutamine (5 g) and BCAA (3 g) on performance (training volume, muscular endurance, muscular strength and anaerobic capacity), blood parameters (albumin, globulin, glucose, electrolytes, hemoglobin, lipid profile, creatinine, urea, etc.) and body composition of individuals submitted to 10 weeks of resistance training. The controversy between these results and those previously mentioned might be due to the different amino acid composition in the supplements offered, resulting in distinct properties of each supplement.

Besides being administered with amino acids, glutamine is also a component of supplements containing several nutrients, such as caffeine and creatine. Gonzalez et al. [57] evaluated the effects of a pre-workout supplement containing glutamine, arginine, leucine, isoleucine, valine, taurine, β-alanine, creatine, glucuronolactone and caffeine (the concentration of each nutrient was not specified), administered 10 min before a resistance training session (four sets of no more than 10 repetitions of barbell squat or bench press at 80% of the 1-repetition maximum–1-RM), to resistance trained men. The authors observed an increase in the number of repetitions, in the average peak and in the mean power performance for all sets when ingesting the pre-workout supplement compared to placebo, but there was no difference between treatments in the reported feelings of energy, focus or fatigue.

Differently, Naclerio et al. [58] compared the administration of a multi-ingredient supplement (containing carbohydrate 53 g, protein 14.5 g, glutamine 5 g, and carnitine 1.5 g) to carbohydrate alone, administered before, during and immediately after a 90-min intermittent repeated sprint test, but did not observe changes in physical performance. Plasma CK concentrations were lower 24 h after exercise when supplementing with multi-ingredient supplement compared to carbohydrate, whereas plasma myoglobin levels were lower 1 h after exercise in carbohydrate trial than placebo. The authors concluded that these interventions do not present an anti-fatigue effect, but can partially attenuate muscle damage.

The same research group, in a similar protocol, verified that this multi-ingredient supplement attenuated fatigue perception without improving performance in soccer players. One hour after the intermittent test, plasma myoglobin levels were lower when administering the multi-ingredient supplement and carbohydrate compared to placebo, whereas carbohydrate supplementation elicited lower neutrophil and monocytes concentrations than multi-ingredient and placebo. There was no difference between trials in other parameters, such as CK, IL-6 and lymphocytes count. The conclusion was similar to the previous study—interventions do not improve performance but may mitigate muscle damage and inflammation induced by physical exercise [59].

Although some of these interventions have presented interesting results, as they contain several nutrients, it is not possible to attribute these effects to any of them, except for their synergetic impact. It is important to emphasize that even in the studies where glutamine was supplemented with several other nutrients, this amino acid was offered in high doses, being, in most of the cases, one of the most prevalent amino acids in the administered supplements.

Furthermore, it is worth highlighting that there are important differences between the studies evaluated, such as the supplementation protocol (dose, supplementation with free glutamine or associated with other nutrients, etc.), the exercise protocol (short-term exercise and aerobic, long-term exercise and endurance or intermittent), characteristics of the volunteers (gender, age, physical activity level, etc.), among others, which might partially explain the controversial results obtained.

The studies mentioned above are shown in Table 1 (human studies) and Table 2 (animal studies).

## 5. Conclusions

The most important findings of the studies evaluated are:Glutamine supplementation seems to increase muscle glycogen synthesis and reduce ammonia accumulation induced by exercise, especially when administered for long-term periods (more than 5 consecutive days). However, concerning glycogen synthesis, more research is needed to establish a greater effect of glutamine in comparison to supplements containing carbohydrate or creatine monohydrate.Glutamine supplementation seems to attenuate markers of muscle damage, such as blood CK and LDH levels.These above-mentioned properties of glutamine are especially interesting for athletes who practice exhaustive and prolonged exercises.Despite improving some fatigue markers, glutamine supplementation seems to have limited effects on physical performance.Supplements containing glutamine associated with several other nutrients seems to present ergogenic effects; nonetheless, it is not possible to attribute these properties to glutamine only.Finally, L-alanyl-L-glutamine supplementation might be used as an alternative to increase glutamine availability. Furthermore, because of its high stability, this dipeptide is a suitable option to be included in commercial products. Nevertheless, it is important highlighting that more research is needed to support the anti-fatigue potential of glutamine supplementation.

## 6. Relevance to Clinical Practice and Limitations

The evaluation of these 55 articles allowed us to discuss the anti-fatigue properties of glutamine and the effects of glutamine supplementation related to exercise-induced fatigue. The results and conclusions obtained in our article may be helpful in clarifying about the anti-fatigue potential of glutamine and guiding on glutamine supplementation in the field of Sports Nutrition.

The main limitation of our article is the reduced number of keywords used in the search (only “glutamine” and “fatigue”). However, our major objective was, indeed, to discuss the anti-fatigue property of glutamine; thus, this limitation did not seem to compromise our aim and neither our results and conclusions.

## Figures and Tables

**Figure 1 nutrients-11-00863-f001:**
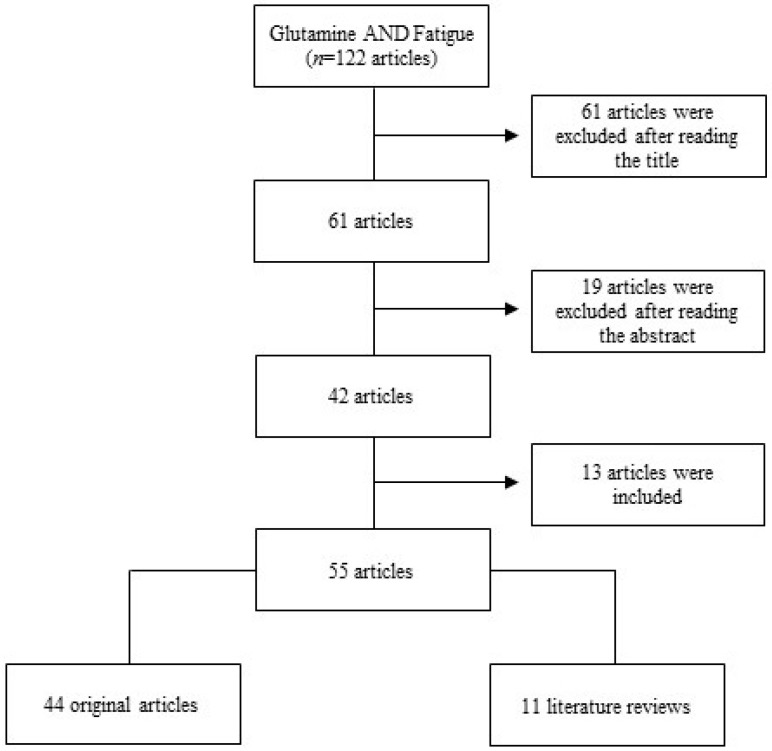
Stages of study—selection and inclusion of articles.

**Figure 2 nutrients-11-00863-f002:**
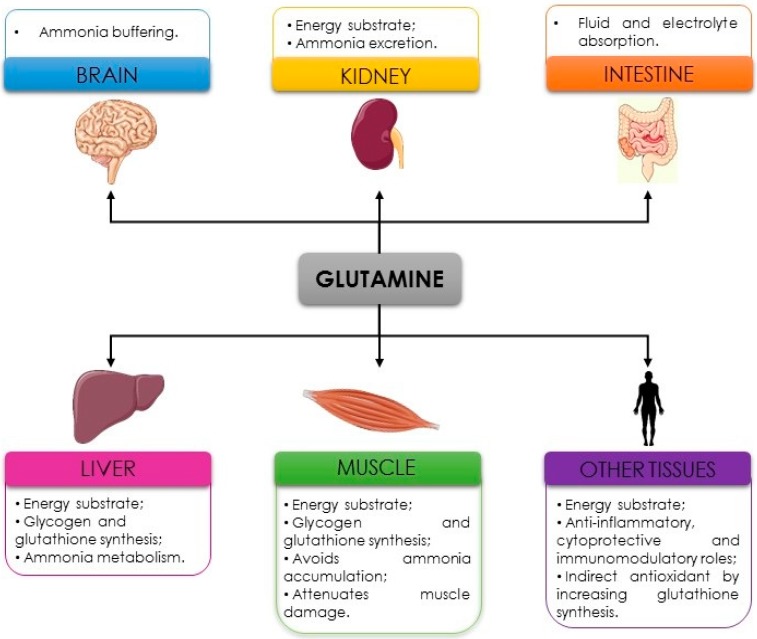
Anti-fatigue properties of glutamine.

**Table 1 nutrients-11-00863-t001:** Human studies involving glutamine administration and fatigue markers (chronological order).

Individuals	Age	Supplementation Protocol	Exercise Protocol	Results	Reference
18 untrained subjects (13 males and 5 females).	17–35 yr	Three infusions after exercise: glutamine (50 mg/kg^−1^/h^−1^), alanine + glycine (30.5 and 25.7 mg/kg^−1^/h^−1^, respectively) and saline (10 mg/kg^−1^/h^−1^).	Cycling at 70–140% of the VO_2max_ for 90 min.	The muscle concentrations of glutamine and glycogen were higher in the subjects treated with glutamine compared to other groups.	Varnier et al. (1995) [16]
7 male subjects.	-	Three drinks after exercise: 18.5% glucose polymer solution, 18.5% glucose polymer solution containing 8 g of glutamine or a placebo containing 8 g of glutamine.	Glycogen-depleting exercise protocol in the cycle ergometer at 70% of the VO_2max_.	Glucose and glutamine solution increased whole body nonoxidative glucose disposal by 25%, whereas oral glutamine alone promoted the storage of muscle glycogen to an extent similar to glucose.	Bowtell et al. (1999) [7]
8 well trained male cyclists.	25 ± 3 yr	Four drinks after exercise: 1—control: 0.8 g/kg of glucose, 2—glutamine: 0.8 g/kg of glucose plus 0.3 g/kg of glutamine, 3—a wheat hydrolysate containing 0.8 g/kg of glucose and 26% of glutamine, and 4—a whey hydrolysate containing 0.8 g/kg of glucose and 6.6% of glutamine.	Intense cycle ergometer exercise.	Supplementation with free glutamine or a carbohydrate mixture containing glutamine did not affect muscle glycogen resynthesis.	van Hall et al. (2000) [43]
Male subjects	-	Glutamine or ornithine α-ketoglutarate both at 0.125 g/kg or placebo.	Bicycle exercise at 70% of the VO_2max_.	Glutamine supplementation increased the muscle concentrations of Krebs cycle intermediates without affecting phosphocreatine depletion, lactate accumulation, and performance.	Rennie et al. (2001) [42]
23 elite rugby players.	27.2 ± 0.4 yr	3.6 g of amino acids (glutamine 0.65 g, leucine, isoleucine, valine, arginine, threonine, lysine, proline, methionine, histidine, phenylalanine and tryptophan) 2 times per day for 90 days.	Rugby.	Supplementation improved reported vigor and earlier recovery from fatigue, as well as increased hemoglobin, red blood cells count, hematocrit and serum iron.	Ohtani et al. (2001) [23]
13 male middle- and long-distance runners.	20.2 ± 0.4 yr	Three different doses of an amino acid mixture (glutamine, leucine, isoleucine, valine, arginine, threonine, lysine, proline, methionine, histidine, phenylalanine and tryptophan): 2.2 g/day for one month, 4.4 g/day for one month and 6.6 g/day for one month.	Sustained exercise (running) for 2–3 h/day, 5 days/week, for 6 months.	Increase in the physical condition score and parameters of oxygen-carrying capacity (hematocrit, hemoglobin, and red blood cells count), and decrease in serum CK after supplementing with the higher dose.	Ohtani et al. (2001) [53]
22 male college students.	19–21 yr	5.6 g of an amino acid mixture (glutamine, leucine, isoleucine, valine, arginine, threonine, lysine, proline, methionine, histidine, phenylalanine and tryptophan), 2 times per day, for 10 days.	One session of eccentric exercise training.	Earlier recovery from muscle fatigue and higher maximum isometric strength in amino acids trial compared to placebo. Moreover, most of the individuals reported less delayed muscle soreness with amino acids supplementation.	Sugita et al. (2003) [54]
13 runners (9 male and 4 female).	18–49 yr	0.1 g/kg of glutamine 4 times a day for 14 days.	Twice-daily interval training for 9–9.5 days.	Increase in nasal IgA concentrations without affecting other immunological parameters and physical performance.	Krieger et al. (2004) [49]
36 resistance-trained males.	31 ± 8 yr	Three supplements for 10 weeks: 1—placebo: 48 g of carbohydrate, 2—40 g of whey protein + 8 g of casein and 3—40 g of whey protein + 3 g of BCAA + 5 g of glutamine.	Resistance training program for 10 weeks.	No effect on physical performance (training volume, muscular endurance, muscular strength, and aerobic capacity), blood parameters and body composition in the group supplemented with glutamine.	Kerksick et al. (2006) [56]
15 male endurance runners.	35.5 ± 9.8 yr	Three supplements: 1—70 mg/kg of glutamine, 2—1 g/kg of sucrose and maltodextrin and 3—glutamine + carbohydrate.	Running for 120 min (~34 km).	Contrary to placebo, there was no increase in blood ammonia in supplemented individuals in the first 30 min of exercise. Additionally, in the last 90 min of running, subjects under supplementation had lower blood ammonia levels compared to placebo.	Carvalho-Peixoto et al. (2007) [44]
18 professional football players.	22.6 ± 0.6 yr	100 mg/kg of glutamine or alanine administered 1 h before exercise (short-term) or for 5 consecutive days (long-term).	Two types of exercise: intermittent (a football match) or with continuous intensity (running for 60 min at 80% of the HR_max_).	Long-term glutamine supplementation protected against hyperammonemia only after intermittent exercise.	Bassini-Cameron et al. (2008) [14]
9 male soccer players.	18.4 ± 1.1 yr	3.5 g of glutamine peptide + 50 g of maltodextrin or only 50 g of maltodextrin administered 30 min before the exercise.	Protocol that simulates the movements of a soccer match (intermittent exercise on the treadmill).	Improvement in the time and distance and reduced feelings of fatigue after supplementation with glutamine peptide and carbohydrate.	Favano et al. (2008) [46]
10 physically active males.	20.8 ± 0.6 yr	L-alanyl-L-glutamine in two doses (0.05 g/kg or 0.2 g/kg) or water.	An exercise session on a cycle ergometer at 75% of the VO_2max_.	Increase in plasma glutamine concentrations with a higher dose of L-alanyl-L-glutamine, as well as increase in the time until exhaustion in both supplemented groups compared to water.	Hoffman et al. (2010) [51]
8 resistance-trained males.	20.6 ± 0.7 yr	Commercial supplement ‘Amino Impact^TM^’, containing 2.05 g of taurine, glucuronolactone, and caffeine, 7.9 g of leucine, isoleucine, valine, arginine and glutamine, 5 g of di-creatine citrate and 2.5 g of β-alanine.	Resistance training session: four sets of no more than 10 repetitions of barbell squat or bench press at 80% of the 1-RM.	Increase in the number of repetitions, in the average peak and in the mean power performance for all sets when ingesting the pre-workout supplement compared to placebo.	Gonzalez et al. (2011) [57]
10 female basketball players.	21.2 ± 1.6 yr	L-alanyl-L-glutamine supplementation in low dose (1 g/500 mL) or high dose (2 g/500 mL) or water (placebo).	40-min basketball game.	Improvement in basketball shooting performance and visual reaction time with a low dose of L-alanyl-L-glutamine compared to water ingestion (placebo).	Hoffman et al. (2012) [41]
16 resistance-trained males.	21 ± 2 yr	Commercial supplement ‘Cyclone^TM^’ containing 30 g of whey protein, 5.1 g of creatine, 5.1 g of glutamine and 1.5 g of HMB, administered 2 times per day, or placebo (maltodextrin), for 12 weeks.	Resistance training—four sessions per week for 12 weeks.	Supplementation did not affect MVIF, time to fatigue at 70% of the MVIF, peak concentric strength and 1-RM of lateral pull. However, cyclone administration increased the number of repetitions for 80% pre-training 1-RM for lateral pull and bench press.	Willems et al. (2012) [55]
28 well-trained males.	20–30 yr	Four supplementations: 1—0.25 g/kg of glutamine, 2—50 g of maltodextrin, 3—glutamine and maltodextrin (0.25 g/kg and 50 g, respectively), and 4—water plus sweetener (placebo).	Running-based anaerobic sprint test, a protocol consisting of 6 times 35 m of discontinuous sprints.	Maximal and minimal power were higher after glutamine and carbohydrate supplementation (together) compared to placebo.	Khorshidi-Hosseini and Nakhostin-Roohi (2013) [47]
Five male elite rowing athletes.	17.2 ± 1.1 yr	Supplementation for 7 days before the test with BCAA (3.15 g/day) or glutamine (6 g/day).	2000 m of rowing at the maximal intensity using an indoor rowing machine.	None of the interventions affected plasma ammonia, lactate, and the cytokines IL-6 and IL-8; nevertheless, glutamine supplementation reduced the plasma levels of CK 30 min after exercise compared to the values measured immediately after training.	Koo et al. (2014) [45]
10 trained males.	25 ± 3.8 yr	Supplementation before, during and immediately after exercise with: 1—multi-ingredient supplement containing 53 g of carbohydrate, 14.5 g of protein, 1.2 g of lipid, 5 g of glutamine and 1.5 g of L-carnitine-L-tartrate, 2—69.5 g of carbohydrate or 3—placebo: low kcal beverage.	90-min intermittent repeated sprint test.	Physical performance did not differ between trials. Plasma CK concentrations were lower 24 h after exercise when supplementing with multi-ingredient supplement compared to carbohydrate, whereas plasma myoglobin levels were lower 1 h after exercise in carbohydrate trial compared to placebo.	Naclerio et al. (2014) [58]
16 male amateur soccer players.	24 ± 3.7 yr	Supplementation before, during and immediately after exercise with: 1—multi-ingredient supplement containing 53 g of carbohydrate, 14.5 g of protein, 1.2 g of lipid, 5 g of glutamine and 1.5 g of L-carnitine-L-tartrate, 2—69.5 g of carbohydrate or 3—placebo: low kcal beverage.	90-m intermittent repeated sprint test.	Multi-ingredient supplement attenuated fatigue perception without improving performance. One hour after the intermittent test, plasma myoglobin levels were lower when administering the multi-ingredient supplement and carbohydrate compared to placebo, whereas carbohydrate supplementation elicited lower neutrophil and monocytes concentrations than multi-ingredient and placebo.	Naclerio et al. (2015) [59]
12 endurance-trained males.	23.5 ± 3.7 yr	Three trials: 1—A sports drink containing 4.9 g of carbohydrate, 113 mg of sodium and 32 mg of potassium with L-alanyl-L-glutamine in two doses (low dose: 300 mg/500 mL or high dose: 1 g/500 mL), 2—only the sports drink (placebo) or 3—without supplementation (no hydration).	One-hour treadmill run at 75% of the VO_2peak_, followed by a run to exhaustion at 90% of the VO_2peak_.	Plasma glutamine was higher and time to exhaustion was longer when supplementing with L-alanyl-L-glutamine compared to the no hydration trial, but there was no difference between L-alanyl-L-glutamine supplementation and the sports drink only (placebo).	McCormack et al. (2015) [52]
11 physically active men (*n* = 7) and women (*n* = 4).	18–44 yr	Supplementation one hour prior to and immediately post exercise with 0.15 g/kg of body weight of glutamine mixed with 2 g of sugar-free lemon drink or only 2 g of sugar-free lemon drink (placebo).	87 min of simulated firefighting exercises (running, shoveling and stepping exercise) in hot conditions (38 °C, 35% relative humidity).	Glutamine supplementation reduced subjective fatigue, ratings of perceived exertion and gastrointestinal damage, besides increasing HSP70 and IκBα in PBMCs.	Nava et al. (2018) [48]

Legend: BCAA: branched-chain amino acids; CK: creatine kinase; HMB: β-hydroxy-β-methylbutyrate; HRmax: maximum heart rate; Ig: immunoglobulin; IκBα: inhibitor of kappa B; IL: interleukin; MVIF: maximum voluntary isometric force; PBMCs: peripheral blood mononuclear cells; RM: repetition maximum; VO2max: maximum oxygen uptake; yr: years.

**Table 2 nutrients-11-00863-t002:** Animal studies involving glutamine administration and fatigue markers (chronological order).

Individuals	Age	Supplementation Protocol	Exercise Protocol	Results	Reference
Adult male rats.	-	A solution containing amino acids (16.6 mmol/L of glutamine, 13.9 mmol/L of arginine, 15.2 mmol/L of leucine, 9.9 mmol/L of isoleucine and 11.1 mmol/L of valine) or water ad libitum.	Exercising on running wheels.	The intake of the amino acids solution reduced the brain release of serotonin (central fatigue marker) and was positively correlated with the volume of exercise.	Smriga et al. (2006) [5]
36 male Wistar rats.	-	Daily dose of 1 g/kg^−1^ of glutamine or 1.5 g/kg^−1^ of L-alanyl-L-glutamine through gavage for 21 days.	Swimming exercise: 60 min/day^−1^, 5 days/week for 6 weeks.	Although supplementations, especially with L-alanyl-L-glutamine, increased glutamine concentrations, there were no differences between groups in the time to exhaustion.	Rogero et al. (2006) [50]
40 male Wistar rats.	2 months	Three supplementations: 1—alanine, 2—free glutamine and alanine, 3—L-alanyl-L-glutamine. Supplements were given in drinking water, diluted to a 4% concentration, and provided ad libitum for 21 days.	Resistance training protocol consisting of climbing a vertical ladder with progressive loads.	Glutamine and alanine supplementation reduced parameters of muscle damage (plasma CK and LDH) and inflammation (plasma TNF-α and IL-1β), and increased anti-inflammatory and cytoprotective markers (plasma IL-6, IL-10, and muscle HSP70), but without improving performance.	Raizel et al. (2016) [17]
40 male Wistar rats.	2 months	Three supplementations: 1—alanine, 2—free glutamine and alanine, 3—L-alanyl-L-glutamine. Supplements were given in drinking water, diluted to a 4% concentration, and provided ad libitum for 21 days.	Resistance training protocol consisting of climbing a vertical ladder with progressive loads.	Glutamine and alanine supplementation reduced the GSSG/GSH ratio in erythrocytes and muscle TBARS, evidencing an antioxidant role, but without improving performance.	Leite et al. (2016) [18]
40 male Wistar rats.	2 months	Three supplementations: 1—alanine, 2—free glutamine and alanine, 3—L-alanyl-L-glutamine. Supplements were given in drinking water, diluted to a 4% concentration, and provided ad libitum for 21 days.	Resistance training protocol consisting of climbing a vertical ladder with progressive loads.	Glutamine and alanine supplementation improved some fatigue markers (reduced muscle ammonia and increased muscle glycogen), but impaired others (increased the free tryptophan/total tryptophan plasma ratio and hypothalamic serotonin concentrations), without affecting performance.	Coqueiro et al. (2018) [33]

Legend: CK: creatine kinase; GSH: glutathione; GSSG: oxidized glutathione; HSP: heat-shock protein; IL: interleukin; LDH: lactate dehydrogenase; TBARS: thiobarbituric acid reactive substances; TNF: tumor necrosis factor.

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
