# Peer review of "Glutamine as an Anti-Fatigue Amino Acid in Sports Nutrition"

_nutrients, 2019, doi:10.3390/nu11040863_

Round 1

Author Response

Reviewer: 1

We are pleased for the careful reading of our manuscript and for your thoughtful comments. We want to extend our appreciation for taking the time and effort necessary to provide an insightful guidance. Your comments, as well as those made by reviewer 2, have been carefully considered.

Abstract

            As suggested, we included a conclusion in the abstract:

“Most of the studies evaluated observed that glutamine supplementation improved some fatigue markers, such as increased glycogen synthesis and reduced ammonia accumulation, but this intervention did not increase physical performance. Thus, despite improving some fatigue parameters, glutamine supplementation seems to have limited effects on performance” (Lines 24 – 27).

Keywords

            As suggested, we excluded the words “glutamine” and “fatigue” and included the words “immune system” and “hydration” (Lines 28 – 29).

Introduction

            As suggested, we changed the sentence “this amino acid is one of the most glycogenic in humans and animals” for “it is one of the most abundant glycogenic amino acids in humans and animals” (Lines 43 – 44).

            As recommended, we changed the word “prevention” to “attenuation” (Line 48).

            As recommended, we changed the sentence “have still not been completely” for “have not been yet completely” (Line 51).

            As suggested, we changed the word “synthesize”, in line 52. We rewrote our objective according to the suggestions made by reviewer 2.

Methods

            As requested, we better explained the term “complete version” by including the sentence “(…) did not provide the complete version of the manuscript (just the abstract)” (Line 73).

Topic: Glutamine and physical exercise

            As requested, we included the unit (g/mol) of glutamine molecular weight (Line 115). 

            As requested, we excluded the word “an” in line 117.

            As suggested, we changed the term “as a consequence” for “therefore” (Line 121).

            As recommended, we changed the word “maintain” for “meet” (Line 142).

            As recommended, we changed the sentence “Decreased glutamine availability is linked with several deleterious effects, such as disorders in the immune system and an increase in the incidence of infections” for “Decreased glutamine availability is linked with disorders in the immune system and an increase in the incidence of infections” in order to exclude the word “deleterious”, which is a strong word to be used in this case (Line 147).

            As requested, we better explained the term “be tolerated” regarding to glutamine supplementation, including in the sentence that it means the lack of side effects: “doses of 20-30 g of glutamine have been reported to be tolerated (no side effects), offering no harm to humans” (Lines 157 – 158).

Topic: Glutamine and its anti-fatigue properties

            As well pointed out by the reviewer, we included the word “mental” in the definition of the term “fatigue”:

“Fatigue is a multiple-cause phenomenon defined as the inability to maintain power output and strength, resulting in physical and mental performance impairment” (Line 165).

As recommended, we excluded the words “and in athletes with overtraining syndrome” in lines 189 and 190.

As well pointed out by the reviewer, we included more information about the study of Kingsbury et al. (1998), explaining why blood glutamine increased after protein intake:

“An increase in protein intake (through lean meat, fish, cheese, milk powder and soya, that is, glutamine-rich foods) to these fatigued athletes enhanced blood glutamine levels and improved physical performance, raising the question about the possible anti-fatigue effects of glutamine supplementation” (Lines 196 – 199).

As suggested, we included the word “abundant” in line 200.

As recommended, we changed the words “this amino acid” for “glutamine” (Line 206).

As requested, we included a reference after the sentence “Ammonia accumulation is an important cause of fatigue since this metabolite is toxic and affects the activity of some flux-generating enzymes, the cell permeability to ions and the ratio of NAD+/NADH [35]” (Line 212).

As requested, we included the word “however” in line 212.

In the sentence: “Glutamine may also attenuate ammonia excess because this amino acid is the main transporter of nitrogen (ammonia) in the body, preventing the muscle accumulation of this metabolite, and favoring ammonia hepatic metabolism, as well as its renal excretion”, in line 222, the reviewer asked if we referred to “endogenous glutamine” or “glutamine supplementation”.

We referred to “endogenous glutamine”. However, since glutamine supplementation increases endogenous glutamine, it may optimize this effect (ammonia transport, avoiding accumulation). Since it can refer directly to “endogenous glutamine” and indirectly to “glutamine supplementation”, we didn’t specify it.

As suggested, we changed the word “excess” for the word “accumulation” in order to better explain the sentence: “Glutamine may also attenuate ammonia accumulation because this amino acid is the main transporter of nitrogen (ammonia) in the body” (Line 222).

As requested, we included the period of glutamine supplementation in this sentence: “Studies in our laboratory showed that glutamine supplementation (for 21 days) reduced the plasma concentrations of creatine kinase (CK) and lactate dehydrogenase (LDH)” (Lines 226 – 227).

As requested, we included a reference in the sentence:

“Several mechanisms might explain this protective effect of glutamine; this amino acid is absorbed through a sodium-dependent transport, increasing the intracellular concentration of sodium ions and promoting water retention, which increases cell hydration and its resistance to lesions [17]” (Lines 228 – 231).

As requested, we rewrote this sentence:

“Moreover, evidence indicates that glutamine in an important donator of glutamate for glutathione synthesis – the most important non-enzymatic antioxidant in the cell – attributing to glutamine an indirect antioxidant effect [18]. Although oxidative stress is considered as a cause of fatigue, it is unclear in the literature whether the increase in glutathione concentrations through glutamine supplementation is capable of delaying fatigue and improving physical performance” (old sentence).

We changed it according to the reviewer’s recommendations:

“Moreover, evidence indicates that glutamine in an important donator of glutamate for glutathione synthesis – the most important non-enzymatic antioxidant in the cell – which may indicate an indirect antioxidant effect of glutamine [18]. Although elevated oxidative stress might contribute to fatigue, it is unclear in the literature whether the increase in glutathione concentrations through glutamine supplementation could attenuate fatigue and improve physical performance” (new sentence) (Lines 234 – 239).

As requested, we excluded the sentence: “It is worth mentioning that limited fluid and electrolyte absorption in the gut affect neural function that commands fine motor control, impairing physical performance” (Lines 249 – 251).

Topic: Effects of glutamine supplementation on exercise-induced fatigue

Subtopic: Glutamine

            We excluded the word “alone” from this topic in order to better represent the studies included in this topic.

            As suggested, we make only one paragraph with the sentences:

“Similarly, Bowtell et al. [7] investigated the effects of glutamine supplementation on whole body carbohydrate storage and muscle glycogen resynthesis in subjects after completing a glycogen-depleting exercise protocol. Individuals cycled on the ergometer at 70% of the VO2max for 30 minutes; thereafter, the workload was doubled and they completed 6 times of 1-minute bursts of activity separated by 2 minutes of rest. Finally, they cycled for 45 minutes at 70% of the VO2max. After exercise, individuals received one of the three drinks: (i) 18.5% glucose polymer solution, (ii) 18.5% glucose polymer solution containing 8 g of glutamine or (iii) a placebo containing 8 g of glutamine”.

And

            “Plasma glucose and insulin were higher when consuming drinks with glucose, and there was a tendency for plasma insulin to be higher after ingesting glucose and glutamine rather than only glucose. Supplementation with glutamine-containing drinks increased plasma glutamine. In the second hour of recovery, glucose and glutamine solution increased whole body nonoxidative glucose disposal by 25%, whereas oral glutamine alone promoted the storage of muscle glycogen to an extent similar to glucose. This result is surprising since it would be expected that the provision of 61 g of glucose polymer (amount of glucose provided in the glucose polymer solution), as opposed to 8 g of glutamine (amount of glutamine provided in the placebo solution), would result in a higher muscle glycogen synthesis; thus, it suggests a great impact of glutamine on muscle glycogen synthesis” (Lines 269 – 286).

            In the sentence: “This result is surprising since it would be expected that the provision of 61 g of glucose polymer, as opposed to 8 g of glutamine, would result in a higher muscle glycogen synthesis; thus, it suggests a great impact of glutamine on muscle glycogen synthesis”, between the lines 282 – 285, the reviewer asked about which group received 61 g of glucose. Therefore, we changed the sentence in order to better explain it:

            “This result is surprising since it would be expected that the provision of 61 g of glucose polymer (amount of glucose provided in the glucose polymer solution), as opposed to 8 g of glutamine (amount of glutamine provided in the placebo solution), would result in a higher muscle glycogen synthesis; thus, it suggests a great impact of glutamine on muscle glycogen synthesis” (Lines 282 – 285).

            As requested, we included the duration (“acute supplementation”) of glutamine supplementation in the study of Rennie et al. (2001) (Line 290).

            As suggested, we included some explanations for the different results between the studies presented: “The different supplementation protocols and administered doses might explain the differences in the results of these studies” (Lines 305 – 306).

            As recommended, we changed the word “glycogen” for “depleted glycogen stores” (Line 307).

            As suggested, we changed the words “feelings of fatigue (Borg’s scale)” for “rate of perceived exertion (RPE)” (Line 331).

            As requested, we changed the word “opposition” for “contrast” (Line 340).

Subtopic: L-alanyl-L-glutamine

            As requested, we changed the word “dietetic” for “dietary” (Lina 346).

            As recommended, we included more information about the intestinal absorption of di- and tripeptides compared to amino acids:

“In order to increase glutamine availability, the supplementation with peptides of glutamine, such as the dipeptide L-alanyl-L-glutamine, has been used, since di- and tripeptides are absorbed across the intestinal epithelium in their intact form by more efficient and faster mechanisms, such as the oligopeptide transporter PepT-1, than free amino acids [17,18,31]. Thus, evidence showed that L-alanyl-L-glutamine supplementation was more effective in increasing plasma, muscle and liver glutamine concentrations compared to free glutamine administration [46]” (Lines 347 – 353). 

As suggested, we excluded the words “on the other hand” in order to avoid the comparison between the study of Rogero et al. (2006), with rats, and Hoffman et al. (2010), with humans (Line 366).

As requested, we included the information “mild dehydration” and “male” in this sentence: “or water to dehydrated male subjects (mild dehydration) submitted to an exercise session on the cycle ergometer” (Lines 367 – 368).

            As requested, we included the measure of exercise performance in the sentence:

“Despite improving several parameters, glutamine and alanine administration did not improve performance evaluated by a maximum carrying capacity test [17,18]” (Lines 396 – 397).

As requested, we better explained the central fatigue hypothesis in this sentence:

“In fact, recently, we observed that these amino acids supplementation improved some fatigue markers, such as muscle ammonia and glycogen, while impaired others, since L-alanyl-L-glutamine administration increased the hypothalamic concentrations of serotonin and the plasma concentrations of its precursor (tryptophan), although without affecting physical performance. It is worth mentioning that serotonin is considered as a parameter of central fatigue, since it is linked to behavioral alterations, such as reduced appetite, sleepiness, and fatigue, reducing the mental and physical efficiency [31]” (Lines 398 – 405).

Subtopic: Glutamine associated with other nutrients

            As requested, we included more information about the amount of glutamine ingested in the study of Ohtani et al. (2001):

“Ohtani et al. [21] observed that an amino acid mixture (glutamine: 0.65 g – the amino acid in the highest concentration in the mixture – leucine, isoleucine, valine, arginine, threonine, lysine, proline, methionine, histidine, phenylalanine and tryptophan) (…)” (Lines 410 – 412).

We have also included information on the amount of glutamine in the other studies presented in this subtopic.

As requested, we included some limitations of the study of Ohtani et al. (2001):

“Some limitations of this study should be highlighted. Firstly, since several amino acids were ingested, it is not possible to attribute the effects to any of them, and, secondly, some of the results (such as reported vigor) were obtained by questionnaires. Thus, several factors could have affected the accuracy of the results” (Lines 417 – 420).

            As suggested, we changed the sentence “as a conclusion, the authors stated” for “the authors concluded that” (Line 444).

            As recommended, we changed the word “synergy” for “synergetic impact” (Line 487).

The reviewer asked whether studies supplementing glutamine with other nutrients should be included in this review. We agree with this concern, however, only a few studies supplemented glutamine alone; the most part supplemented glutamine with other nutrients, such as carbohydrate, L-alanine (L-alanyl-L-glutamine), and other amino acids. Thus, if we decided to exclude the trials where glutamine was supplemented with another nutrient, there would be very few studies, insufficient to create a literature review. We would like to emphasize that even in the studies where glutamine was supplemented with several other nutrients, this amino acid was offered in high doses, being, in most of the cases, one of the most prevalent amino acids in the administered supplements.

Table 1

            As requested, we separated the animal studies from human studies. Thus, we created two tables – Table 1 (human studies) and Table 2 (animal studies).

Conclusions

            As requested, we specified the term “long-term periods”. The information included was supported by Bassini-Cameron et al. (2008), who observed that 5-day-supplementation with glutamine was more effective in reducing blood ammonia than acute glutamine supplementation:

“Glutamine supplementation seems to increase muscle glycogen synthesis and reduce ammonia accumulation induced by exercise, especially when administered for long-term periods (more than 5 consecutive days)” (Lines 507 – 509).

            The reviewer asked why most of our conclusions are not specific to the anti-fatigue effect of glutamine. We agree with this observation; nonetheless, based on the results from the studies evaluated, we included the effects of glutamine supplementation on fatigue markers (the ones that may compromise performance), such as glycogen, ammonia and muscle damage parameters. We also included that glutamine supplementation seems to have limited effects on physical performance, since most of the studies evaluated did not find changes in physical performance (through different tests, such as exhaustion tests and maximum carrying capacity tests) after glutamine supplementation.  

Reviewer 2 Report

Authors aimed to perform an integrative review of glutamine as an anti-fatigue supplement in sports. However, the manuscript presents several issues regarding methodology and the real applications of this dietary supplement are arguable based on the current evidence.

Abstract: Please describe in the abstract how many publications met the inclusion criteria and were analyzed throughout the text.

Line 19: It seems that the paper focus on both main mechanisms of action and effects of glutamine supplementation; however, there is a continous combination of concepts and any deep analysis from a biochemical point of view to qualify this as "mechanisms of action" (i.e., authors do not mention any signaling pathway which can elucidate the physiological effects). 

Line 47: Please define the problem and the objective of the review. In the title authors refers exclusively to anti-fatigue properties to glutamine supplementation; in the last comment (line 19) they aimed to review mechanisms of action and effects after supplementing; finally, at the end of introduction authors aims to synthesize evidence regarding anti-fatigue and ergogenic properties.

Lines 48-51: Methodology is poor. Since this review belongs to the category of integrative review, authors should based their framework analysis on the five stages employed by Whittemore and Knafl (2005) and the improvement by Hopia et al. (2016). 

Lines 52-55: Authors should describe in detail the methodology sections for a well-structured review (integrative review methodology, search strategy, inclusion/exclusion criteria and data analysis). In order to strenghten the work, authors should include a risk of bias analysis (e.g., Cochrane).

Lines 56-61: All this paragraph should be included in the results section.

Line 107: Check this section. 0.65 g per kg is equivalent to 45,5 g for a 70 kg subject. The range mentioned by authors in the same line goes from 20 to 30 g.

Line 113: Even though some molecular mechanisms are described by authors, there is not deep analysis in the mechanisms itself. This seems more an analysis of physiological response to glutamine administration. 

Lines 187-198: Given that many of the general analysis developed by authors comes from animal studies, and considering the recent position stands of well-recognized organizations in the field (such as, ISSN, AIS, IOC) where glutamine has been displaced and marked as non-effective / little to no evidence of efficacy, authors should be cautious when refer to positive effects on performance. Here is important to define if continue discussing ergogenic effects.

Please check the following: 

ISSN = https://doi.org/10.1186/s12970-018-0242-y

IOC = http://dx.doi.org/10.1136/bjsports-2018-099027

AIS = ABCD Classification update

I recommend that authors make emphasis on the specific population of athletes where glutamine supplementation might have a positive effect (please define if this positive effect will be in terms of anti-fatigue only or ergogenic also). For example, a good population target is ultra-endurance athletes.

Line 199: Title of this section is fuzzy. Please consider to change.

Lines 222-224: This can not be established from a single study. Many other studies have failed showing this effect. Actually, there is not doubt that other molecules (i.e., creatine monohydrate) impact in a greater degree the glycogen synthesis.

Line 279: In previous paragraphs authors mentioned no sinergy but in this line highlight this point. Please check.

Line 367-389: It is important to mention the glutamine concentrations / doses in those formulations (specially because table 1 do not specify the amount of glutamine in those mult-ingredient products). Also, authors should be cautious that the positive effects when supplementing with amino acids formulas are more due to the combination of them but not by glutamine itself.

Line 416-418: If anti-fatigue and erogenic properties are analyzed in the same table, authors should divide in corresponding sections within the table 1. Also, do not forget to include a footer to describe all acronyms that appear in the table.

Conclusions: Analysis section missing... Conclusions are OK but authors should note that focusing on those sports with better potential to include this substance (such as, ultra-endurance athletes) might have a better impact in sports nutrition community rather than repeating what has been established by previous organizations in regards to glutamine supplementation.

Author Response

Reviewer: 2

Comments to the Author

Authors aimed to perform an integrative review of glutamine as an anti-fatigue supplement in sports. However, the manuscript presents several issues regarding methodology and the real applications of this dietary supplement are arguable based on the current evidence.

We sincerely appreciate the careful and thorough reading of our manuscript and would like to thank the constructive suggestions raised. We agree with all the requested changes, which are presented below.

Abstract

Please describe in the abstract how many publications met the inclusion criteria and were analyzed throughout the text.

We agree with the issue pointed out and, for this reason, we included this information in the abstract:

 “The study was carried out in the PubMed database, using the keywords combination “glutamine” and “fatigue”. Fifty-five studies met the inclusion criteria and were evaluated in this literature review” (Lines 22 – 24).

Line 19: It seems that the paper focus on both main mechanisms of action and effects of glutamine supplementation; however, there is a continuous combination of concepts and any deep analysis from a biochemical point of view to qualify this as "mechanisms of action" (i.e., authors do not mention any signaling pathway which can elucidate the physiological effects).

We comprehend this point of view; nonetheless, in our article, we pointed out (in detail) the main properties of glutamine that could attenuate fatigue markers (those that contribute to fatigue development and performance impairment), such as glycogen synthesis, ammonia detox, attenuation of muscle damage and oxidative stress, among others. In order to better explain our proposal, we changed the words “mechanisms of action” for “properties” throughout the manuscript (Lines 19; 163; 253; 256).

Line 47: Please define the problem and the objective of the review. In the title authors refers exclusively to anti-fatigue properties to glutamine supplementation; in the last comment (line 19) they aimed to review mechanisms of action and effects after supplementing; finally, at the end of introduction authors aims to synthesize evidence regarding anti-fatigue and ergogenic properties.

We agree with the issue pointed out. Our objectives are to review the anti-fatigue properties of glutamine (as requested, we changed the words “mechanisms of action”) and the effects of glutamine supplementation regarding this subject (fatigue). We included this information in the introduction, as requested:

“Despite the potential of glutamine in attenuating some causes of fatigue, the effects of this amino acid supplementation on fatigue markers and physical performance have not been yet completely elucidated. Thus, the present article aims to review the main anti-fatigue properties of glutamine and the effects of glutamine supplementation in the context of fatigue induced by physical exercise” (Lines 50 – 55).

Lines 48-51: Methodology is poor. Since this review belongs to the category of integrative review, authors should based their framework analysis on the five stages employed by Whittemore and Knafl (2005) and the improvement by Hopia et al. (2016).

Lines 52-55: Authors should describe in detail the methodology sections for a well-structured review (integrative review methodology, search strategy, inclusion/exclusion criteria and data analysis). In order to strenghten the work, authors should include a risk of bias analysis (e.g., Cochrane).

We agree with the issue pointed out and we really appreciate the suggestion, thus, we changed the topic methodology, as requested. However, we would like to emphasize that our article is not a systematic review, for this reason, some of the requests were not possible to make:

Search methods and exclusion criteria

The study was carried out in the PubMed database in February 2019, using the descriptor Mesh (Medical Subject Headings), without limitation as to the publication period. The keywords combination used was “Glutamine” AND “Fatigue” (n=122 articles).

Articles discussing fatigue related to diseases or that included animals or humans with any stated medical condition were excluded from this study. Only the articles that addressed the relationship between glutamine and fatigue induced by physical exercise were included in this review. Additionally, unpublished manuscripts (such as dissertations and thesis) were not included in this review.

Data analysis

One hundred and twenty-two articles were found, including experimental studies with animals and humans, and literature reviews. After reading the title of these studies, 61 articles were excluded, since they had no correlation with the subject (effects of glutamine supplementation on exercise-induced fatigue) or did not provide the complete version of the manuscript (just the abstract). Of the 61 articles that remained, 19 articles were excluded after reading the abstract, since they did not have a correlation with the theme, remaining 42 studies.

After reading the complete version of these 42 selected articles, 13 other studies, which were cited in the articles evaluated, but were not obtained in the search, were included, totalizing 55 articles – 44 original studies and 11 literature reviews (Figure 1).

Presentation stage

Fifty-five articles, which evaluated and/or discussed glutamine supplementation, alone or associated with other nutrients, in the context of fatigue induced by physical exercise, were included in this review.

Concerning the animal and human studies, the aspects of all of these articles were described thoroughly. Certain features of these studies, such as author, participants, study design and findings were described in tables. Moreover, the limitations of these studies were discussed.

Strengths and limitations of this review

The evaluation of these 55 articles allowed us to discuss the anti-fatigue properties of glutamine and the effects of glutamine supplementation related to exercise-induced fatigue. The results and conclusions obtained in our article may be helpful in clarifying about the anti-fatigue potential of glutamine and guiding on glutamine supplementation in the field of Sports Nutrition.

The main limitation of our article is the reduced number of keywords used in the search (only “glutamine” and “fatigue”). However, our major objective was, indeed, discuss the anti-fatigue property of glutamine; thus, this limitation did not seem to compromise our aim and neither our results and conclusions” (Lines 59 – 112).

Lines 56-61: All this paragraph should be included in the results section.

This paragraph was included in the section “data analysis” after modifications in the topic “methods”:

Data analysis

One hundred and twenty-two articles were found, including experimental studies with animals and humans, and literature reviews. After reading the title of these studies, 61 articles were excluded, since they had no correlation with the subject (effects of glutamine supplementation on exercise-induced fatigue) or did not provide the complete version of the manuscript (just the abstract). Of the 61 articles that remained, 19 articles were excluded after reading the abstract, since they did not have a correlation with the theme, remaining 42 studies.

After reading the complete version of these 42 selected articles, 13 other studies, which were cited in the articles evaluated, but were not obtained in the search, were included, totalizing 55 articles – 44 original studies and 11 literature reviews (Figure 1)” (Lines 69 – 78).

Line 107: Check this section. 0.65 g per kg is equivalent to 45,5 g for a 70 kg subject. The range mentioned by authors in the same line goes from 20 to 30 g.

We agree with this issue, for this reason, we excluded the information “0.65 g per kg”:

“Moreover, doses of 20-30 g of glutamine have been reported to be tolerated” (Line 157).

Lines 187-198: Given that many of the general analysis developed by authors comes from animal studies, and considering the recent position stands of well-recognized organizations in the field (such as, ISSN, AIS, IOC) where glutamine has been displaced and marked as non-effective / little to no evidence of efficacy, authors should be cautious when refer to positive effects on performance. Here is important to define if continue discussing ergogenic effects.

We agree with the issue pointed out and, as requested, we included an observation about it:

“It is important mentioning that some of these results (attenuation of muscle damage and oxidative stress parameters) were obtained from animal studies, thus, it is not possible to guarantee that the same effects would occur in human trials” (Lines 240 – 242). 

I recommend that authors make emphasis on the specific population of athletes where glutamine supplementation might have a positive effect (please define if this positive effect will be in terms of anti-fatigue only or ergogenic also). For example, a good population target is ultra-endurance athletes.

We agree with this issue, then, we included this information:

“Considering the properties presented, glutamine seems to be an interesting supplement for fatigue attenuation, especially for athletes who practice endurance sports (exhaustive and prolonged exercise)” (Lines 251 – 253).

Line 199: Title of this section is fuzzy. Please consider to change.

As requested, we changed the title “Glutamine administration as an anti-fatigue intervention: Studies over the years” for “Effects of glutamine supplementation on exercise-induced fatigue” (Line 258).

Lines 222-224: This cannot be established from a single study. Many other studies have failed showing this effect. Actually, there is not doubt that other molecules (i.e., creatine monohydrate) impact in a greater degree the glycogen synthesis.

We agree with this issue; thus, we included this information:

“However, no other study has shown that the impact of glutamine supplementation on glycogen synthesis is the same as the one promoted by glucose” (Lines 286 – 287).

Line 279: In previous paragraphs authors mentioned no sinergy but in this line highlight this point. Please check.

As requested, we included information about it:

“These data suggest that the combination of glutamine and carbohydrate is more efficient in preventing anaerobic power decrease and increasing performance than glutamine alone, emphasizing the synergy between glutamine and carbohydrate, although some studies did not corroborate this finding” (Lines 341 – 344).

Line 367-389: It is important to mention the glutamine concentrations / doses in those formulations (specially because table 1 do not specify the amount of glutamine in those mult-ingredient products). Also, authors should be cautious that the positive effects when supplementing with amino acids formulas are more due to the combination of them but not by glutamine itself.

We agree with this consideration and we already included this information in the text and in the tables.

As requested, we included a footer describing all of the acronyms that appear in the tables.

Conclusions: Conclusions are OK but authors should note that focusing on those sports with better potential to include this substance (such as, ultra-endurance athletes) might have a better impact in sports nutrition community rather than repeating what has been established by previous organizations in regards to glutamine supplementation.

We agree with the issue pointed out; thus, we included this sentence in the topic Conclusions:

“These above-mentioned properties of glutamine are especially interesting for athletes who practice exhaustive and prolonged exercises” (Lines 512 – 513).

Round 2

Reviewer 1 Report

Dear authors,

Thank you for addressing my comments and answering my questions! Just one last comment regarding your answer to the question below. 

"We would like to emphasize that even in the studies where glutamine was supplemented with several other nutrients, this amino acid was offered in high doses, being, in most of the cases, one of the most prevalent amino acids in the administered supplements. We would like to emphasize that even in the studies where glutamine was supplemented with several other nutrients, this amino acid was offered in high doses, being, in most of the cases, one of the most prevalent amino acids in the administered supplements."

I would recommend you adding the highlighted part to the manuscript where you talk about multi-ingredients supplements.  

Author Response

Reviewer: 1

Dear authors,

Thank you for addressing my comments and answering my questions! Just one last comment regarding your answer to the question below:

“We would like to emphasize that even in the studies where glutamine was supplemented with several other nutrients, this amino acid was offered in high doses, being, in most of the cases, one of the most prevalent amino acids in the administered supplements. We would like to emphasize that even in the studies where glutamine was supplemented with several other nutrients, this amino acid was offered in high doses, being, in most of the cases, one of the most prevalent amino acids in the administered supplements”.

I would recommend you adding the highlighted part to the manuscript where you talk about multi-ingredients supplements. 

We agree and appreciate for your suggestion. We included this information between lines 490 – 492.  

Reviewer 2 Report

COMMENTS:

Line 22: It's better to express this idea in terms of: "PubMed database was selected to examine the literature,..."

Line 24: Please consider this review as an integrative review. Authors should based their framework analysis on the five stages employed by Whittemore and Knafl (2005) and the improvement by Hopia et al. (2016). This process does not take to much time and improves the scientific soundness of the manuscript.

Line 53: English editing. Repetition of glutamine in same line, please check.

Line 53-54: You can remove "... in the context of fatigue induced by physical exercise" and improve redaction, maybe including at the end "in this regard".

Line 57: Systematic review is not the only one in which this approach is performed. In fact, authors should refer to Whittemore and Knafl (2005) recommendations for this kind of review.

Line 60-61: It's better to express this idea in terms of: "PubMed database was selected to examine the literature,...". Also, february without capital letter and first mention "Medical Subject Headings" before "MeSH terms". 

Line 69: Please rename this section. This is more suitable for results or data extraction. 

Line 70: If you already filtered articles to accomplish inclusion criteria, why do you mention animal studies in this line as an outcome? Please, just refer to the articles of the first outcome (after using PubMed filters). If you decided to include animal studies, you should make clearer this in the inclusion criteria.

Line 73: Remove "(just the abstract)".

Line 76-78: This approach of combining experimental and theoretical publications is typical of an Integrative Review.

Line 94: Please use the in-arrow to report that 13 extra-article were included. Consider this flow diagram according to the PRISMA guidelines.

Line 96: Please rename this section. This is more suitable for "data synthesis".

Line 104-112: Please relocate this section. It should be after conclusions.

Line 240-242: Please check English editing. Why not mentioning ISSN, IOC and AIS position stands?

Line 253: Please refer as potential properties or effects, since more research is needed.

Line 286: Check redaction. You can refer to "However, there is limited evidence that reproduce this effects on glycogen synthesis in athletes population."

Line 507-509: Authors should add: "...; however, more research is needed to stablish a greater effect of glutamine in comparison to supplements containing carbohydrate or creatine monohydrate".

Line 518 (item 6): Before recommending to include this glutamine form, authors should recall the importance of more research in athletes to support this claims.

Line 523: Do not forget to declare funding.

Author Response

Reviewer: 2

Line 22: It's better to express this idea in terms of: "PubMed database was selected to examine the literature..."

            We agree and appreciate your consideration. As suggested, we changed the sentence for:

“PubMed database was selected to examine the literature, using the keywords combination ‘glutamine’ and ‘fatigue’” (Lines 22 – 23).

Line 24: Please consider this review as an integrative review. Authors should based their framework analysis on the five stages employed by Whittemore and Knafl (2005) and the improvement by Hopia et al. (2016). This process does not take to much time and improves the scientific soundness of the manuscript.

            We agree with the issue pointed out. As requested, we changed the term “literature review” for “integrative literature review” (Line 24). We also rewrote our article (especially the methods) according to the five stages proposed by Whittemore and Knafl (2005) and the improvement by Hopia et al. (2016) (Lines 58 – 111; Lines 530 – 538).

Line 53: English editing. Repetition of glutamine in same line, please check.

            We apologize for this mistake. As suggested, we changed the sentence:

“Thus, the present article aims to review the main anti-fatigue properties of glutamine and the effects of this amino acid supplementation in this regard” (Lines 53 – 54).

Line 53-54: You can remove “... in the context of fatigue induced by physical exercise” and improve redaction, maybe including at the end “in this regard”.

As recommended, we changed the sentence for:

“Thus, the present article aims to review the main anti-fatigue properties of glutamine and the effects of this amino acid supplementation in this regard” (Lines 53 – 54).

Line 57: Systematic review is not the only one in which this approach is performed. In fact, authors should refer to Whittemore and Knafl (2005) recommendations for this kind of review.

            As suggested, we also rewrote our article (especially the methods) according to the five stages proposed by Whittemore and Knafl (2005) and the improvement by Hopia et al. (2016) (Lines 58 – 111; Lines 530 – 538).

Line 60-61: It's better to express this idea in terms of: "PubMed database was selected to examine the literature,...". Also, february without capital letter and first mention "Medical Subject Headings" before "MeSH terms".

            According to the suggestions, we changed the sentence:

“PubMed database was selected to examine the literature, in february 2019, using the descriptor Medical Subject Headings (MeSH), without limitation as to the publication period. The keywords combination used was “Glutamine” AND “Fatigue” (n=122 articles)” (Lines 69 – 71).

Line 69: Please rename this section. This is more suitable for results or data extraction.

            As suggested, we changed the title of the section for “data extraction” (Line 78).

Line 70: If you already filtered articles to accomplish inclusion criteria, why do you mention animal studies in this line as an outcome? Please, just refer to the articles of the first outcome (after using PubMed filters). If you decided to include animal studies, you should make clearer this in the inclusion criteria.

            We excluded the sentence “including experimental studies with animals and humans, and literature reviews” (Line 79).

Line 73: Remove "(just the abstract)".

            Since the inclusion of the sentence: “did not provide the complete version of the manuscript (just the abstract)” was a suggestion given by reviewer 1, we cannot change it.

Line 96: Please rename this section. This is more suitable for "data synthesis".

            As recommended, we changed the title of the section for “data synthesis” (Line 104).

Line 104-112: Please relocate this section. It should be after conclusions.

            As suggested, we included this section after the conclusions (Lines 530 – 538).

Line 240-242: Please check English editing. Why not mentioning ISSN, IOC and AIS position stands?

            As requested, we included information about the position stands of these organizations:

“In addition, recent position stands of well-recognized organizations, such as the International Society of Sports Nutrition (ISSN) and the International Olympic Committee (IOC), have considered glutamine as a non-effective supplement, with little or no evidence of efficacy (Kerksick et al., 2018; Maughan et al., 2018)” (Lines 241 – 244).

Line 253: Please refer as potential properties or effects, since more research is needed.

            As requested, we changed the word “properties” for “potential properties” (Line 253).  

Line 286: Check redaction. You can refer to "However, there is limited evidence that reproduce this effects on glycogen synthesis in athlete’s population."

            As suggested, we changed the sentence for:

“However, there is limited evidence concerning this effect on glycogen synthesis in the athlete population” (Line 288 – 289).

Line 507-509: Authors should add: "...; however, more research is needed to stablish a greater effect of glutamine in comparison to supplements containing carbohydrate or creatine monohydrate".

            As requested, we included the sentence:

“However, concerning glycogen synthesis, more research is needed to stablish a greater effect of glutamine in comparison to supplements containing carbohydrate or creatine monohydrate” (Lines 514 – 516).

Line 518 (item 6): Before recommending to include this glutamine form, authors should recall the importance of more research in athletes to support this claim.

            We agree with the issue pointed out; thus, we included the sentence:

“Nevertheless, it is important highlighting that more research is needed to support the anti-fatigue potential of glutamine supplementation” (Lines 527 – 528).

Line 523: Do not forget to declare funding.

            We appreciate the reminder. As requested, we included the funding (Lines 542 – 544). 
